# Sarcoid-Like Reaction in Non-Hodgkin’s Lymphoma—A Diagnostic Challenge for Deauville Scoring on ^18^F-FDG PET/CT Imaging

**DOI:** 10.3390/diagnostics11061009

**Published:** 2021-05-31

**Authors:** Michael Winkelmann, Kai Rejeski, Marion Subklewe, Jens Ricke, Marcus Unterrainer, Martina Rudelius, Wolfgang G. Kunz

**Affiliations:** 1Department of Radiology, University Hospital, LMU Munich, 81377 Munich, Germany; michael.winkelmann@med.uni-muenchen.de (M.W.); jens.ricke@med.uni-muenchen.de (J.R.); marcus.unterrainer@med.uni-muenchen.de (M.U.); 2Department of Medicine III, University Hospital, LMU Munich, 81377 Munich, Germany; kai.rejeski@med.uni-muenchen.de (K.R.); marion.subklewe@med.uni-muenchen.de (M.S.); 3Department of Pathology, University Hospital, LMU Munich, 81377 Munich, Germany; martina.rudelius@med.uni-muenchen.de

**Keywords:** lymphoma, NHL, sarcoid-like reaction, PET/CT, pitfall, Deauville

## Abstract

The sarcoid-like reaction represents an autoinflammatory cause of mediastinal and hilar lymphadenopathy but may also involve other lymph node regions and organs. This rare phenomenon has mainly been reported in patients with Hodgkin’s lymphoma (HL) or solid tumors (particularly melanoma) undergoing immunotherapy and chemotherapy. Cases in non-Hodgkin’s lymphoma (NHL) are very uncommon. We present an uncommon case of a patient with primarily mediastinal diffuse large B-cell lymphoma (DLBCL) who showed a CT-based partial response in interim staging, whereas at end-of-treatment multiple newly enlarged and hypermetabolic mediastinal and bilateral hilar lymph nodes were detected by ^18^F-FDG PET/CT imaging. A subsequent histological workup determined a sarcoid-like reaction without any lymphomatous tissue. Therefore, sarcoid-like reactions should be considered as a potential pitfall in Deauville staging with ^18^F-FDG PET/CT imaging for patients with NHL.

The sarcoid-like reaction represents an autoinflammatory cause of mediastinal and hilar lymphadenopathy [1,2]. This rare phenomenon has mainly been reported in patients with solid tumors undergoing immunotherapy (particularly in those with melanoma), but can also be seen in patients treated with chemotherapy [3,4,5,6]. In addition to the involvement of regional lymph nodes, a sarcoid-like reaction can also affect distant lymph node regions or organs such as the lung, spleen, bone marrow, and skin [7,8]. Rare cases of sarcoid-like reactions have been reported in patients with HL [9,10,11]. Cases of sarcoid-like reactions in patients with NHL are less common [10,11,12]. Some authors propose the concept of a lymphoma-sarcoidosis syndrome, in which patients have a coexistence of sarcoidosis and lymphoma, particularly HL [13]. A study comparing the prognosis of patients presenting with DLBCL with and without sarcoidosis showed no significant difference in overall survival and progression-free survival between the two groups [14].

In this case (Figure 1), the occurrence of the newly appearing hypermetabolic mediastinal and bilateral hilar lymph nodes could have been misinterpreted as a Deauville score of 5 (DS 5). Lymphoma patients with a DS 5 show a poor prognosis and have a higher chance of relapse compared to a DS 4 [15,16].

Overall, this case underlines the importance of the tissue biopsy of unclear newly appearing lesions in patients with lymphoma. In clinical routine, sarcoid-like reactions should be considered as a potential pitfall in Deauville staging with ^18^F-FDG PET/CT imaging, even in NHL patients and in cases with high metabolic activity.

## Figures and Tables

**Figure 1 diagnostics-11-01009-f001:**
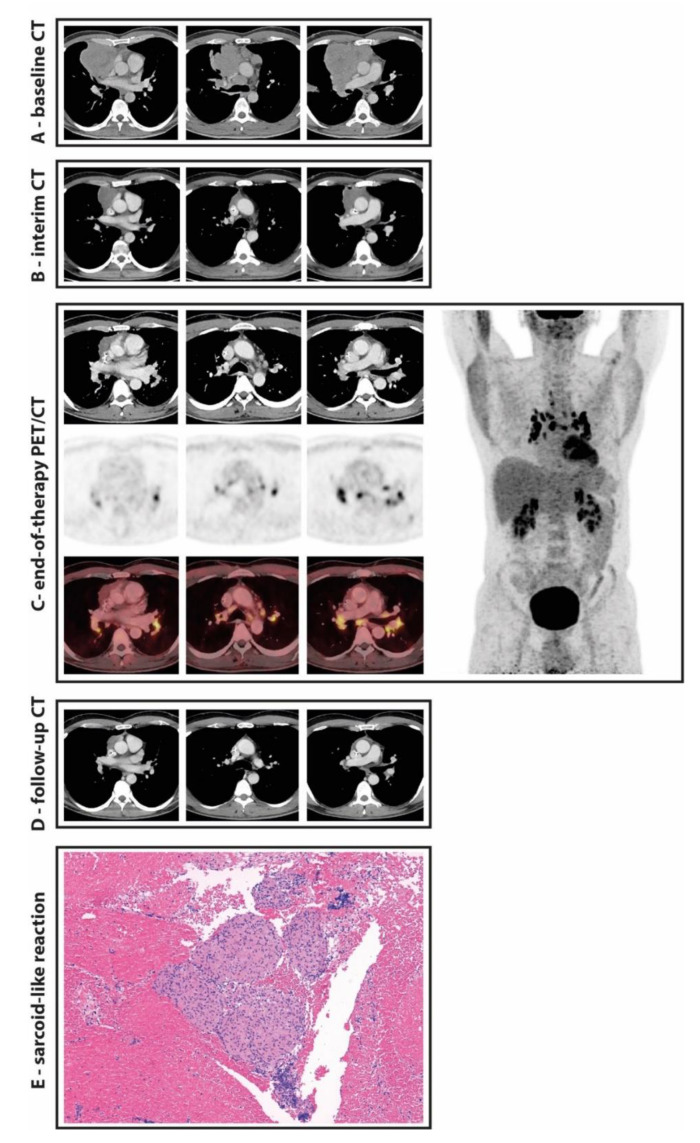
A 55-year-old male patient with primarily mediastinal diffuse large B-cell lymphoma (DLBCL; **A**) underwent 6 cycles of chemotherapy and 2 cycles of rituximab (R-CHOP-14). At interim staging after 3 cycles of therapy, CT-based partial response was achieved (**B**). After completion of therapy (**C**), staging was carried out using 18F-FDG PET/CT imaging for an additional assessment of metabolic response. The pre-existing mediastinal lymphoma manifestation continued to decrease significantly in size (**C**) in addition to a complete metabolic response (Deauville 2). At the same time, however, multiple newly enlarged and hypermetabolic mediastinal and bilateral hilar lymph nodes (SUVmax 17) were detected in locations previously unaffected by the DLBCL (**C**). These findings triggered a biopsy with endobronchial ultrasound and transbronchial needle aspiration. The following histological workup determined multiple epithelioid cell granulomas without any necrosis or lymphomatous tissue, which was compatible with a sarcoid-like reaction (**E**). The subsequent CT staging 3 months later without any further therapy showed a normalization in size of the mediastinal and bilateral hilar lymph nodes (**D**).

## Data Availability

Not applicable.

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
