# Peer review of "Sarcoid-Like Reaction in Non-Hodgkin’s Lymphoma—A Diagnostic Challenge for Deauville Scoring on 18F-FDG PET/CT Imaging"

_diagnostics, 2021, doi:10.3390/diagnostics11061009_

Round 1

Reviewer 1 Report

This is a concise presentation of a rare case of sarcoid-like reaction mimicking active lymphoma on end-of- therapy FDG PET/CT.

The presentation is well written and clearly emphasizes the clinicaly important awareness to such false positive findings and the critical need for histologic confirmation of such positive PET/CT findings, presenting at unexpected sequence and/or locations.

Very few comments related to data presentation and typos, should be addressed:

  1. Abstract and last paragraph on lines 35-37, should state more accurately the facts that sarcoid-like reaction may appear in many sites not only in mediastinal or hilar nodes, and that it may appear in realtion to chemotherapy as well.
  2. Images and Figure legend:

2a. typo in the age of the patient, line 24 (it is not " figure 55"…)

2b. typo in the first word on line 34 ("and").

2c. The first "C" reference should appear immediately after the words "After completion of therapy" since "C" presents both CT and PET (line 26).

2d. The term "follow-up" should not be used for all the different stages of patient management. The title of the images therefore should be changed to: A- baseline CT, B- interim CT, C- end-of-therapy PET/CT, D- follow up CT

2e. In order to be more representative and convincing, the middle image at "follow up 3" should be at a slightly lower level, so that the left pre-vascular and bilateral para-tracheal lymph nodes at the level of the carina are clearly shown and can be compared to the prior middle image.

3. It is interesting to know whether a repeated PET/CT was performed on follow up of this patient at some time point. It is well known that reduction in size of lymph nodes does not necessarily indicate resolution of metabolic activity.

Reviewer 2 Report

I find it a pity that the authors present PET/CT images after completion of therapy and not before as well, so that a comparison could be made.

Although the content of this article (in quantity) is mainly images, as indeed should be the case for the “Interesting Images” section of the Journal, the main text is even smaller than the abstract (I counted 94 words of text in the main article, while there seem to be 119 in the abstract)!

The case presented is indeed rare (combination of NHL and sarcoidosis), but other similar case reports have already been published elsewhere before (https://www.ncbi.nlm.nih.gov/pmc/articles/PMC4855092/ as an example). Actually, Chalayer et al. in 2015 report several such cases (https://pubmed.ncbi.nlm.nih.gov/25660608/). 

Reviewer 3 Report

The interesting image presented by Winkelmann et al. shows the occurrence of a sarcoid-like reaction in Non-Hodgkin Lymphoma treated with R-CHOP.

The case presented is somewhat interesting, as it displays a rare pitfall in FDG PET/CT image interpretation. However, I propose a few comments about its presentation and discussion:

1) In the figure, it is not clear why the authors presented the histological finding (Panel E) close to the staging CT scan. Consider moving it together with Panel C.

2) The clinical description and the discussion are insufficient. The authors state that this case represents a challenge for the Deauville Scoring system. I generally agree with this assumption. Indeed, in primary mediastinal B-cell lymphoma DS 4 is generally associated with a good prognosis (probably due to inflammation in the large residual mediastinal masses typical of this disease), while patients with DS 5 have a high chance of relapse. In this case, the occurrence of a sarcoid-like reaction could be misinterpreted as DS 5. However, these considerations should be clearly stated in the discussion to help this message to be addressed to a wider readership.  
